# Trace Element Imbalances in Acquired Hepatocerebral Degeneration and Changes after Liver Transplant

**DOI:** 10.3390/biology12060804

**Published:** 2023-05-31

**Authors:** Henrique Nascimento, Maria João Malaquias, Catarina Mendes Pinto, José Sá Silva, Dina Rochate, Cristina Fraga, José Eduardo Alves, Cristina Ramos, Judit Gandara, Sofia Ferreira, Vítor Lopes, Sara Cavaco, Helena Pessegueiro Miranda, Agostinho Almeida, Marina Magalhães

**Affiliations:** 1Neurology Service, Centro Hospitalar Universitário de Santo António, 4099-001 Porto, Portugal; 2Neuroradiology Service, Centro Hospitalar Universitário de Santo António, 4099-001 Porto, Portugal; 3Hematology Service, Hospital do Divino Espírito Santo, 9500-370 Ponta Delgada, Portugal; 4Hepatic Pancreatic Transplantation Unit, Centro Hospitalar Universitário de Santo António, 4099-001 Porto, Portugal; 5Neuropsychology Unit, Centro Hospitalar Universitário de Santo António, 4099-001 Porto, Portugal; 6Laboratório Associado para a Química Verde (Associated Laboratory for Green Chemistry) of the Network of Chemistry and Technology (LAQV/REQUIMTE), Department of Chemical Sciences, Faculdade de Farmácia, Universidade do Porto, 4050-313 Porto, Portugal

**Keywords:** acquired hepatocerebral degeneration, chronic liver disease, liver transplantation, trace elements, manganese

## Abstract

**Simple Summary:**

The liver is responsible for the metabolism and elimination of many deleterious substances from the human body. Consequently, toxic substances can accumulate in patients with chronic liver disease, particularly in those with surgically or spontaneously induced portosystemic shunts. So-called acquired hepatocerebral degeneration (AHD) is associated with the accumulation of manganese, a trace element, in the brain of patients with liver disease. The symptoms include movement disorders and cognitive impairment. Liver transplantation (LT) is the only available treatment. In this study, we evaluated the blood levels of trace elements in patients with AHD before and after LT and compared them with those of healthy controls. The association between trace element levels and changes in the neurological examination was also assessed. The impact of LT on clinical and analytical variables was evaluated in a subgroup of patients. We found significantly different levels of many trace elements, and they were generally higher in AHD patients compared to those of the controls, although they were still within normal reference ranges. Some trace elements linked to oxidative stress and inflammation (e.g., zinc, copper and selenium) were particularly affected in patients, suggesting a putative role in the pathophysiology and symptomatology of AHD. The neurological symptoms and trace element imbalances were at least partially reversible after LT.

**Abstract:**

Brain manganese (Mn) accumulation is a key feature in patients with acquired hepatocerebral degeneration (AHD). The role of trace elements other than Mn in AHD needs to be clarified. In this study, using inductively coupled plasma mass spectrometry, we aimed to evaluate blood levels of trace elements in patients with AHD before and after liver transplantation (LT). Trace element levels in the AHD group were also compared with those of healthy controls (blood donors, n = 51). Fifty-one AHD patients were included in the study (mean age: 59.2 ± 10.6 years; men: 72.5%). AHD patients had higher levels of Mn, Li, B, Ni, As, Sr, Mo, Cd, Sb, Tl and Pb and a higher Cu/Se ratio, and lower levels of Se and Rb. Six patients (two women; mean age 55 ± 8.7 years) underwent LT, and there was an improvement in neurological symptoms, a significant increase in the Zn, Se and Sr levels, and a decrease in the Cu/Zn and Cu/Se ratios. In summary, several trace element imbalances were identified in AHD patients. Liver transplantation resulted in the improvement of neurological manifestations and the oxidant/inflammatory status. It is possible that observed changes in trace element levels may play a role in the pathophysiology and symptomatology of AHD.

## 1. Introduction

Trace elements play an important role in biological systems, and both their deficiency and excess can have deleterious effects. The different elements have distinct deficiency/ toxicity profiles and may affect more significantly a particular organ/system [1].

Acquired hepatocerebral degeneration (AHD) is a neurological syndrome that develops in patients with chronic liver disease (CLD), e.g., hepatic cirrhosis, and is associated with the accumulation of Mn in the central nervous system (CNS) [2]. The prevalence of AHD in CLD patients is estimated to be 1–2% [1]. The clinical features of AHD include movement disorders, primarily parkinsonism and ataxia-plus syndrome, as well as cognitive impairment with psychiatric features [2].

Two of the most important mechanisms involved in the accumulation of toxic substances in CLD include the reduced metabolic function of the failing liver and the development of a portosystemic shunt that bypasses the organ and allows toxic substances to reach other tissues and accumulate there [1]. 

The accumulation of Mn in the CNS is associated with the appearance of bilateral hyperintense areas in the globus pallidus on T1-weighted magnetic resonance images (MRI), especially in patients with parkinsonism. In turn, patients with ataxia-plus syndrome may also show high-signal lesions in the middle cerebellar peduncles on T2-weighted images [1,2].

In addition to Mn, other trace elements tend to accumulate in the presence of liver disease, particularly those whose excretion mainly occurs via the hepatobiliary route. However, the impact of CLD on one’s overall physiology is considerable, and its influence is broader than just involving the elements that pass through hepatic metabolism [3,4]. Trace elements play an important role in many biochemical and physiological processes in biological systems, e.g., as cofactors of several key enzymes. Furthermore, their optimal levels are usually within a very narrow range, and either an excess or a deficiency can cause deleterious effects [5].

Acquired hepatocerebral degeneration is a chronic progressive disease without spontaneous recovery. The only available treatment option is liver transplantation (LT). Patients who received LT show an improvement in neurologic symptoms, mainly motor and cognitive impairments, in addition to presenting neuroradiological evidence of a reduction in Mn deposition in the CNS [2,6,7,8]. Therefore, AHD is a formal indication for LT [9].

This study aimed to: (1) evaluate the total blood levels of a wide panel of trace elements (Mn, Ag, As, B, Be, Bi, Cd, Co, Cs, Cu, In, Li, Mg, Mo, Ni, Pb, Rb, Se, Sr, Sn, Sb, Te, Tl and Zn) in patients with AHD and compare them with those of a control group of healthy individuals (blood donors); (2) describe the relationship between factors, such as the etiology of CLD, markers of CLD, and abnormalities in the neurological and neuropsychological examinations in trace elements levels; (3) evaluate clinical and analytical changes after LT.

The levels of various trace elements in AHD patients differed significantly from those of the control group, with a worse antioxidant status being particularly noticeable in AHD patients. Liver transplantation was followed by an improvement in neurological symptoms and a reduction in oxidative stress markers.

## 2. Materials and Methods

### 2.1. Patients

The study included patients with CLD and a working diagnosis of AHD who were referred to our tertiary movement disorders center between 2008 and 2021. AHD was diagnosed according to the following criteria: (1) clinical neurological manifestations; (2) globus pallidus hyperintensity on T1-weighted brain MRI; (3) CLD [1].

All patients underwent a standard protocol of observation and investigation, including neurological and neuropsychological assessments, examinations and evaluations related to hepatic disease, the determination of blood trace element levels and MRI. Patients who underwent LT were re-evaluated after approximately 6 months.

A control group consisting of Portuguese blood donors was used for the comparison of blood trace element levels.

The study protocol was approved by the Ethics Committee of Centro Hospitalar Universitário de Santo António. (Ref 2021.279 (225-DEFI/233-CE).)

### 2.2. Neurological Examination

A complete neurological examination of all patients was performed, and the following findings were recorded as being present or absent: cognitive impairment; behavioral changes; mood disorders; dysarthria; parkinsonism; tremor (resting, postural or myoclonic; unilateral or bilateral); dystonia; gait disorder; oculomotor dysfunction; peripheral neuropathy; pyramidal signs; cerebellar syndrome. Parkinsonism was defined as bradykinesia in association with a tremor and/or rigidity. Neurological disease duration was defined as the time between the onset of neurological manifestations and the baseline neurological evaluation.

### 2.3. Neuropsychological Evaluation

The Dementia Rating Scale-2 (DRS-2) was applied to measure overall cognitive function. Test scores were adjusted to age and education according to normative data for the Portuguese population. Cognitive impairment was assumed when a performance was below the 5th percentile of the demographically adjusted value.

### 2.4. Hepatic Evaluation

All patients were followed up at a hepatology or pre-transplantation outpatient clinic. Clinical evaluation, the laboratory study, the radiological study and/or liver biopsy were previously performed to establish the diagnosis and etiology of CLD. Portal hypertension was defined according to the presence of typical manifestations (e.g., splenomegaly, variceal hemorrhage and ascites), together with findings acquired via abdominal ultrasonography or intravenous contrast computed tomography. The Child–Pugh scale and the Model for End-stage Liver Disease plus serum sodium (MELD-Na) score were used to assess liver disease severity [10,11]. Hepatic disease duration was defined as the time between the diagnosis of CLD and the baseline neurological evaluation.

### 2.5. Laboratory Evaluation

Blood levels of trace elements were determined from whole blood samples collected into BD Vacutainer^®^ Trace Element tubes (K_2_EDTA) (Franklin Lakes, NJ, USA) and analyzed via inductively coupled plasma mass spectrometry (ICP-MS). The elemental isotopes ^7^Li, ^9^Be, ^11^B, ^25^Mg, ^55^Mn, ^59^Co, ^60^Ni, ^65^Cu, ^66^Zn, ^75^As, ^82^Se, ^85^Rb, ^88^Sr, ^98^Mo, ^107^Ag, ^111^Cd, ^115^In, ^118^Sn, ^121^Sb, ^125^Te, ^133^Cs, ^205^Tl, ^206^Pb, ^207^Pb, ^208^Pb and ^209^Bi were measured for analytical determinations, and ^89^Y, ^141^Pr and ^159^Tb were monitored as internal standards (IS). A complete description of the analytical procedure and operational parameters of the ICP-MS instrument, as well as the results obtained in the analytical quality control, can be found in Azevedo et al. [12]. Ammonia was measured using an enzymatic spectrophotometric method (reference range: 26–47 µmol/L). The median of all available ammonia values for the 6-month period before and after the baseline neurological examination was calculated to deal with the large fluctuation in blood ammonia levels.

### 2.6. Statistical Analysis

Statistical analysis was performed using IBM SPSS Statistics, version 27 (SPSS Inc., Chicago, IL, USA). Normally distributed variables are presented as mean ± standard deviation (SD); variables not normally distributed are presented as median (interquartile range). The Mann–Whitney U-test was used for group comparisons at the baseline. Comparisons between time periods were made using the Wilcoxon test. Spearman rank correlation was used to assess correlations between variables. Categorical variables distributions were compared using the χ^2^ test. Statistical significance was defined at *p* < 0.05 using a two-sided test.

## 3. Results

### 3.1. Cross-Sectional Study

Fifty-one patients with AHD were included in the study. A summary of demographic, clinical, neurological and analytical data is presented in Table 1.

The baseline levels of trace elements in AHD patients were compared to a gender-matched control group of blood donors (n = 51; men: 72.5%). The results are summarized in Table 2. The AHD patients’ group was significantly older than the controls were (mean ± SD: 59.2 ± 10.6 years vs. 44.0 ± 8.3 years; *p* < 0.001). A large percentage of Ag, Be, Bi, Sn and Te results were below the limit of detection (LOD) (respectively, 0.282, 0.031, 0.019, 0.131 and 0.157 µg/L); so, these elements were excluded from the comparison between groups. Most subjects, both in the control and patient groups, had trace element levels within the normal range. In addition to higher levels of Mn, patients also had significantly higher levels of Li, B, Ni, As, Sr, Mo, Cd, Sb, Tl and Pb and a higher Cu/Se ratio, and lower levels of Se and Rb. No differences were found for Mg, Co, Cu, Zn, In and Cs. 

Patients with AHD were evaluated according to gender. When baseline variables were compared between subgroups, men had a higher percentage of alcoholic liver disease (89.2% vs. 21.4%, *p* < 0.001), higher serum ammonia (82.3 µmol/L (64.0–101.0) vs. 44.4 µmol/L (31.4–80.0), *p* = 0.022), higher Rb (3.45 mg/L (2.85–4.52) vs. 2.68 mg/L (2.34–3.31), *p* = 0.031) and lower Mo (0.90 µg/L (0.72–1.17) vs. 1.10 µg/L (0.95–1.61), *p* = 0.038) levels. Lower levels of In in men were of borderline significance (0.064 µg/L (0.044–0.074) vs. 0.73 µg/L (0.062–0.082), *p* = 0.077). No other differences were found between genders, namely regarding hepatic disease severity, neurological examination or other trace elements. 

The impact of liver disease severity on different variables of AHD patients was evaluated by dividing the group according to the Child–Pugh classification. This classification divides CLD patients into three categories of worsening hepatic impairment, from A to C, and resulted in n = 21 (A), n = 20 (B) and n = 10 (C) (Table 3). No differences were found between these categories regarding age, gender, hepatic disease-related variables (except for an expected increase in MELD from A to C, reflecting increased liver failure) and neurological examination. It should be noted that only four patients had no history of hepatic encephalopathy (all Child–Pugh category A, *p* = 0.064). Arsenic, B, Co, Cs, Li and Tl levels tended to increase with hepatic dysfunction from Child–Pugh A to C, reaching significance when some of the categories were compared. Lead followed the same pattern, but the difference between subgroups was of borderline significance. On the contrary, the amount of Se decreased significantly with worsening liver disease. No differences were found for the other trace elements.

When patients were evaluated for alcoholic (n = 36) or non-alcoholic (n = 15) hepatic disease, it was found that patients with liver disease of alcoholic etiology had much higher blood Pb levels (35.1 µg/L (16.7–53.0) vs. 11.1 µg/L (7.8–22.1), *p* = 0.003) and borderline higher levels of ammonia (82.1 µmol/L (63.0–98.5) vs. 44.4 µmol/L (28.9–96.3), *p* = 0.058). In addition to the already described higher prevalence of hepatic disease of alcoholic etiology among men, no other difference was found between patients with or without alcoholic liver disease, namely with regard to age, hepatic and neurological variables and other trace elements.

After dividing the AHD patients group according to whether or not they had any specific alteration in the neurological examination, we evaluated their relationship with the clinical, hepatic disease-related variables and trace element data. The results of the evaluation of these associations can be summarized as follows:Patients with cognitive decline (n = 46) showed no significant difference with the cognitively preserved ones. Only borderline higher levels of Ni (1.36 μg/L (1.07–1.64) vs. 1.14 μg/L (0.56–1.37), *p* = 0.078) and Sr (20.7 μg/L (14.9–25.3) vs. 15.0 μg/L (11.0–18.8), *p* = 0.084) were noted.Patients with behavioral disorders (n = 9) had lower As levels (10.35 μg/L (8.43–11.12) vs. 13.92 μg/L (10.06–24.98), *p* = 0.016).Patients with mood disorders (n = 11) had higher Mn ((18.62 μg/L (12.32–29.85) vs. 12.46 μg/L (8.58–19.92), *p* = 0.039) and Mg levels (44.8 mg/L (38.3–52.0) vs. 37.1 mg/L (31.7–44.0), *p* = 0.025).Dysarthria (n = 14) was borderline associated with a shorter CLD duration (2.5 years (1.6–5.8) vs. 4.3 years (2.7–7.6), *p* = 0.079) and higher ammonia (93.5 µmol/L (71.6–102.4) vs. 67.9 µmol (38.1–93.2), *p* = 0.057) and Cd levels (0.444 μg/L (0.342–0.766) vs. 0.361 μg/L (0.197–0.546) *p* = 0.072).Parkinsonism (n = 30) was associated with a shorter neurologic disease duration (1.8 years (1.1–2.9) vs. 3.3 years (1.6–5.2), *p* = 0.034), lower Cs levels (3.23 μg/L (2.23–4.04) vs. 4.33 μg/L (2.95–4.97), *p* = 0.011) and borderline lower Mg (37.8 mg/L (32.0–42.9) vs. 42.6 mg/L (35.5–49.0), *p* = 0.080), Ni (1.22 μg/L (0.90–1.58) vs. 1.39 μg/L (1.26–1.65), *p* = 0.077) and Tl levels (0.066 μg/L (0.043–0.115) vs. 0.081 μg/L (0.066–0.146), *p* = 0.051).Patients with a tremor (n = 34) had a higher prevalence of hepatic encephalopathy (100% vs. 76.5%, *p* = 0.01).Dystonia (n = 4) was associated with a longer CLD duration (13.6 years (4.8–23.9) vs. 3.5 years (2.2–7.0), *p* = 0.034), higher In levels (0.080 μg/L (0.074–0.091) vs. 0.065 μg/L (0.046–0.076), *p* = 0.024) and borderline higher Zn levels (9.80 mg/L (6.39–10.3) vs. 4.92 mg/L (4.11–9.68), *p* = 0.097) and a lower Cu/Zn ratio (0.122 (0.073–0.134) vs. 0.181 (0.125–0.220), *p* = 0.053).Patients with a gait disorder (n = 32) had lower Cd levels (0.362 μg/L (0.188–0.478) vs. 0.450 μg/L (0.306–0.772), *p* = 0.032) and borderline higher Se levels (130 μg/L (120–162) vs. 121 μg/L (115–134), *p* = 0.082).Oculomotor changes (n = 11) were associated with higher ammonia (93.2 µmol/L (73.5–101.9) vs. 67.2 µmol/L (38.0–93.3), *p* = 0.033), Mn (21.88 μg/L (17.61–28.08) vs. 11.28 μg/L (9.13–17.06), *p* = 0.003) and Se levels (148 μg/L (128–199) vs. 124 μg/L (116–142), *p* = 0.023), and borderline higher Mg (44.7 mg/L (36.9–54.2) vs. 38.3 mg/L (31.9–44.0), *p* = 0.051) and Ni levels (1.61 μg/L (1.38–1.82) vs. 1.30 μg/L (1.02–1.54), *p* = 0.052).Peripheral neuropathy (n = 5) was associated with higher Zn levels (11.32 mg/L (7.19–12.21) vs. 4.92 mg/L (4.09–9.66), *p* = 0.038), and a borderline longer neurological disease duration (3.8 years (2.4–6.4) vs. 1.9 years (1.2–3.4), *p* = 0.086), higher Pb levels (43.65 μg/L (30.17–73.57) vs. 25.74 μg/L (11.34–46.88), *p* = 0.097) and a lower Cu/Zn ratio (0.124 (0.102–0.156) vs. 0.180 (0.129–0.220), *p* = 0.061).Pyramidal changes (n = 8) were associated with lower Cs levels (2.22 μg/L (1.75–3.75) vs. 3.88 μg/L (2.51–4.56), *p* = 0.046) and a borderline lower age (56.8 years (49.3–59.7) vs. 60.9 years (54.3 vs. 66.4), *p* = 0.064).When AHD patients were divided according to the presence of cognitive fluctuations (n = 41) and cerebellar changes (n = 11), there were no differences between the subgroups.

Spearman’s correlation was used to assess the relationship between variables. When the entire set of individuals studied was considered (n = 102), age correlated positively with Mn (r = 0.419, *p* < 0.001), Li (r = 0.638, *p* < 0.001), B (r = 0.402, *p* < 0.001), Ni (r = 0.298, *p* = 0.002), Cu (r = 0.309, *p* = 0.002), As (r = 0.572, *p* < 0.001), Sr (r = 0.380, *p* < 0.001), Mo (r = 0.505, *p* < 0.001), Cd (r = 0.267, *p* < 0.001), Sb (r = 0.268, *p* = 0.006), Tl (r = 0.448, *p* < 0.001) and Pb (r = 0.467, *p* < 0.001) and the Cu/Se ratio (r = 0.478, *p* < 0.001) and negatively with Se (r = −0.300, *p* < 0.002) and Rb (r = −0.479, *p* < 0.001). When AHD patients and controls were evaluated separately, significant correlations with age were maintained only for Li, B, As, Sr and Pb in the controls and only for Cu in the AHD patients group.

In AHD patients, correlations between hepatic disease duration, neurological disease duration, MELD, ammonia and trace element levels were also tested (Table 4). The most relevant results were:Hepatic disease duration did not correlate with any other variable.The MELD score was negatively correlated with neurological disease duration and Se levels, while it was positively correlated with higher Co and Cd levels.Ammonia was positively correlated with Mn and negatively correlated with Li.Manganese levels were positively correlated with Mg, Co, Ni, Cu, Rb, In and Sr and negatively correlated with Cu/Se ratio.The Cu/Zn ratio was negatively correlated with Mg, Ni, Se, Rb, Sb, Cs and Tl.The Cu/Se ratio was positively correlated with Mo.

### 3.2. Longitudinal Study

Nine of the AHD patients underwent LT. Two died of peri-transplantation complications, and one did not attend the post-LT evaluation. The remaining six patients were evaluated before and after LT (55 years (52.1–60.8); two (33%) women; CLD duration 4.0 years (1.6–11.7) years; four (66.7%) had alcoholic liver disease; MELD 14 (10–21), Child–Pugh A, B and C categories, with two patients each; AHD duration 1.2 years (0.4–3.6)). Patients who underwent LT were not significantly different from the others in terms of any baseline clinical or laboratory variables. 

Many trace elements showed significant variation with LT, including an increase in Zn, Se, Sr and Pb levels and a decrease in the Cu/Zn and Cu/Se ratios. Manganese levels decreased as well, although without statistical significance (Table 5). Ammonia levels also decreased significantly (83.1 µmol/L (58.4–99.4) vs. 28.4 µmol/L (11.4–33.4), *p* = 0.028)). The median time between LT and laboratory evaluation was 170 days, ranging from 88 to 399 days.

Despite there being an overall trend of patients’ improvement regarding changes in the neurological assessment, only parkinsonism reached statistical significance, with the four patients who had it no longer presenting with it after LT (*p* = 0.046). No other statistically significant changes were observed for clinical neurological variables (Table 5). The median time between LT and neurological evaluation was 222 days (range: 193–333 days). 

Patients also tended to improve in neuropsychological assessments after LT, but again, the difference did not reach statistical significance. The median time between LT and the neuropsychological evaluation was 190 days (range: 175–274 days).

T1-weighted MRI hyperintensity in the globus pallidus disappeared after LT in four of six patients (*p* = 0.046). The median time between LT and MRI was 223 days (range: 191–350 days).

## 4. Discussion

The AHD patients group had a median CLD duration of 3.8 years (2.3–7.4), while neurological symptoms were present for at least 2.2 years (1.3–3.7). The time span between the diagnosis of hepatic disease and the development of AHD is surprisingly wide, ranging from 1 to 33 years [13]. Thus, other variables, such as CLD severity, the degree of portosystemic shunt and age at onset, among others, possibly modulate the development of AHD [1]. On the other hand, the etiology of CLD does not seem to influence the likelihood of developing AHD [1,14].

When they were compared to a gender-matched control group, AHD patients showed significant differences in the levels of many of the trace elements analyzed. Most trace elements presented higher levels in AHD patients (Li, B, Ni, As, Sr, Mo, Cd, Sb, Tl and Pb and the Cu/Se ratio), with only Se and Rb having lower blood concentrations. In any case, all elements were within the widely accepted normal range, which is an interesting finding. Antimony was the exception, with the levels being well above the reference interval, but this was also observed in the controls. The increased Sb levels must be attributed to the blood collection tubes (Sb is commonly present in plastics), as recently observed by Yang et al. [15], who highlighted that special care has to be taken in the determination of this trace element to avoid acquiring falsely elevated results. Regardless, we assumed that all samples were subject to the same bias, and we decided to keep the Sb results, with this special caveat regarding their interpretation. 

When all study subjects were considered, a correlation with age was noticeable with many of the trace elements. Since the control group was significantly younger than the AHD patient group was, age could have acted as a confounding factor, and there might be some overlap of the impact of CLD with age-related changes on trace element levels. However, it appears that the underlying disease, rather than age, is the main factor causing the observed differences, since when correlations with age were separately evaluated in the controls and patients, most of them lost statistical significance, remaining only for Li, B, As, Sr and Pb in the controls and Cu in the AHD patients. It should be emphasized that the duration of neither hepatic disease nor neurological diseases were found to be significantly correlated with any trace element, and, therefore, they are unlikely to be confounding variables in this analysis.

Cirrhosis is more common among men [16,17], which justifies the fact that most of our AHD patients were men (72.5%). Furthermore, some authors have reported that the being a man may itself be a risk factor for AHD [1,18]. The higher percentage of alcohol-related cirrhosis among men (89.2% vs. 21.4% among women) was also in line with a higher prevalence of alcoholism among men [16]. Gender-related differences in AHD patients were present only for increased ammonia and Rb and decreased Mo levels. This highlights that the CLD metabolic impact likely outweighs the gender-associated differences. In agreement, we found no other difference between men and women in the AHD group regarding other hepatic and neurological variables.

When patients were subdivided based on the severity of liver disease, As, B, Co, Cs, Li and Tl levels were increased, while the Se level was decreased, according to the impairment of liver function. Selenium levels have been reported to decrease in direct association with liver disease severity [19,20]. It is important to note that although AHD patients usually have moderate-to-severe CLD (Child–Pugh categories B and C), this is not the only factor influencing the AHD severity, as the degree of portosystemic shunting is also an important contributing factor [1,14].

In addition to a higher proportion of men, patients with CLD of alcoholic etiology also had higher Pb levels. Increased blood concentrations of Pb have been associated with the consumption of alcoholic beverages, particularly wine [21]. This association may somehow explain the Pb levels found in our cohort, since the Portuguese population have a high consumption of alcohol per capita, namely wine [22,23]. Thus, a predominantly alcoholic etiology of liver disease may have contributed to the trace element profile found, in particular, the increased Pb and the decreased Se levels in patients with AHD [24,25]. 

We further divided the study group based on the presence or absence of changes in the neurological examination and found that several trace elements had significantly different levels between the resulting subgroups. Apart from the well-established association between Mn and AHD [1,2], data on the possible contribution of other trace elements to the pathophysiology and clinical manifestations of this disease are scarce. For most of the differences found, a putative underlying mechanism is unclear, and proposing a possible explanation would be unwise. In fact, although some of the associations found may have pathophysiological relevance, in other cases, they may be completely accidental. Looking at the changes in trace element levels between the pre- and post-LT periods may be more helpful in clarifying which elements are actually affected by CLD, and therefore, could potentially contribute to the clinical manifestations of AHD. 

Liver transplantations are known to improve neurological symptoms and decrease brain Mn deposits in AHD patients [2,8]. Of the six patients with complete pre- and post-LT data in our study, four presented with parkinsonism at the baseline. In all of them, the clinical manifestations of parkinsonism were gone after transplantation. The globus pallidus T1-weighted hyperintensity also disappeared in four out of six patients after LT. Concomitantly, the blood Mn levels also decreased, although the change did not reach statistical significance. Other studies have reported that blood Mn level is not the best marker of AHD severity, as it does not have a perfect association with clinical features and brain deposits [2]. 

After LT, the AHD patients showed an increase in Zn and Se levels and a decrease in the corresponding Cu/Zn and Cu/Se ratios. These variations are in agreement with the literature [26,27] and have been associated with an improvement in the general anti-oxidant/anti-inflammatory status [28,29]. However, the magnitude of the contribution of an improved oxidant status to post-LT clinical improvement is unknown. Changes in the levels of some trace elements, such as Cu, Se and Zn, may contribute to oxidative stress via reducing the activity of antioxidant enzymes (e.g., glutathione peroxidase and superoxide dismutase) or by inducing the formation of reactive oxygen species and free radicals [25,28].

Low serum Zn levels in patients with liver cirrhosis are associated with decreased intake, absorption and bioavailability, in addition to increased losses and use [4]. In addition to its involvement in the antioxidant system, Zn also plays an important role in the CNS, being involved in the regulation of long-term potentiation and synaptic plasticity. Thus, Zn deficiency has been associated with disturbances in learning, memory and emotional stability [30]. 

Higher Cu levels in CLD can be explained by increased absorption, decreased excretion by the failing liver and tissue breakdown with the subsequent release of Cu stores [4]. Excess Cu tends to accumulate in the liver and other tissues, including the brain, causing hepatic and neurological symptoms. Wilson’s disease, a condition caused by excessive Cu accumulation, is associated with tremor, parkinsonism, dystonia, dysarthria, gait abnormalities and cognitive impairment [31]. Considering the neurological consequences of excess Cu, the similarity of its manifestations to those of AHD and its close association with liver disease, it is reasonable to consider the contribution of Cu accumulation to AHD. Increased levels of Cu and decreased levels of Zn are associated with a pro-oxidant status. Actually, the Cu/Zn ratio has been reported to be clinically more important than the concentration of any of these trace metals alone are, as it is a sensitive marker of oxidative stress [28].

The decrease in Se levels observed in CLD is probably multifactorial and involves factors, such as decreased intake, impaired absorption and the direct consequences of liver damage. In fact, the impaired hepatic synthesis of Se-containing proteins, notably selenoprotein P, which accounts for more than 50% of plasma Se, contributes to low Se levels in CLD patients [25,26]. These changes are associated with a worsening of the antioxidant status, which has been described as underlying neurodegeneration. In fact, the decreased function of selenoenzymes in the CNS has been reported as a possible contributor to the pathophysiology of disorders such as Alzheimer’s disease and Parkinson’s disease [32]. Importantly, LT can lead to an improvement in Se levels and a recovering Se status after surgery is associated with a positive prognosis [26].

Another potential contribution to a worse antioxidant status in AHD is CLD-related increased ammonia levels, which also improve after LT. Hyperammonemia can activate nitric oxide synthase signal transduction, causing nitrosative stress in the CNS and further contributing to oxidative stress and pro-inflammatory changes in CLD [33]. Consistent with previous studies, our patients showed significantly lower ammonia levels after LT [2].

Lead is an element with a long list of harmful effects, which include central and peripheral nervous system toxicity [21]. With regard to cirrhotic patients, Pb can potentially contribute to carcinogenesis. The mechanism probably includes oxidative stress, as hepatocellular carcinoma cells have shown higher Pb levels and a worse oxidative status as compared to those of healthy liver tissue [34]. The decrease in blood Pb concentration after LT is probably a combined consequence of the surgical procedure and alcohol abstinence, a pre-LT recommendation (minimum of 6 months). Considering the variation in the time interval between blood collection for trace element analysis and transplantation (median of 6 months and 19 days (range: from 22 days to 22 months)), it is difficult to unravel the contributions of each factor.

The relationship between Sr and liver disease is less well known. The post-LT increase in blood Sr levels may be related to mobilization from bone tissue, the body’s main store of Sr [35]. In fact, early bone loss has been described after LT and is attributed to immunosuppressive agents [36]. Thus, increased bone turnover could contribute to the higher Sr values found after LT. The role of Sr in the brain is unknown. A possible role in neurotransmission has been suggested, as Sr may replace Ca in motor endplates and stimulate acetylcholine release [37].

This study has some limitations that must be pointed out. The sample size is relatively small, especially when one is considering its longitudinal component. The fact that the AHD patients were significantly older than the control group were makes it difficult to completely separate changes in trace element levels associated with the disease from those related to age. The short follow-up period makes it impossible to detect possible changes in trace elements and neurological variables that may take longer to develop. Finally, we measured all the elements in whole blood samples when some of them are more commonly determined in other biological specimens (e.g., serum/plasma or urine); so, well-defined reference values were not always available. However, this last limitation was partially eliminated with the use of a control group.

## 5. Conclusions

To the best of our knowledge, this is the first study conducted to address trace element imbalances other than Mn in AHD and how these imbalances are affected by LT. Other trace elements (besides Mn) likely contribute to the physiopathology and clinical manifestations of AHD via implication in the perpetuation of oxidative stress and inflammation in the brains of those patients. Consequently, the improvement in oxidative status could at least partially explain the clinical, analytical and imaging improvements in AHD patients following LT. However, the exact biochemical mechanisms underlying most of the associations found still need to be clarified. More studies are needed to further elucidate the potential role of each individual element and oxidative stress/inflammation in AHD.

## Figures and Tables

**Table 1 biology-12-00804-t001:** Demographic and clinical data of acquired hepatocerebral degeneration (AHD) patients (n = 51).

Demographic Characteristics	
Men (%)	37 (72.5%)
Age (years), mean ± SD	59.2 ± 10.6
Hepatic disease	
CLD duration (years), median (P25–P75)	3.8 (2.3–7.4)
Alcoholic etiology, n (%)	36 (70.6%)
MELD, median (P25–P75)	12 (10–16)
Child–Pugh, n (%)	A: 21 (41.2%) B: 20 (39.2%) C: 10 (19.6%)
Portal hypertension, n (%)	51 (100%)
Hepatic encephalopathy, n (%)	47 (92.2%)
Ammonia (µmol/L); median (P25–P75)	74.4 (43.2–96.5)
Neurological evaluation	
CLD duration (years), median (P25–P75)	3.8 (2.3–7.4)
ND duration (years), median (P25–P75)	2.2 (1.3–3.7)
Cognitive decline, n (%)	46 (90.2%)
Behavioral disorders, n (%)	9 (17.6%)
Mood disorders, n (%)	11 (21.6%)
Dysarthria, n (%)	14 (27.5%)
Parkinsonism, n (%)	30 (58.8%)
Tremor, n (%)	34 (66.7%)
Dystonia, n (%)	4 (7.8%)
Gait disorders, n (%)	32 (62.7%)
OCM dysfunction, n (%)	11 (21.6%)
Peripheral neuropathy, n (%)	5 (9.8%)
Pyramidal symptoms, n (%)	8 (15.7%)
Cerebellar syndrome, n (%)	11 (21.6%)
Neuropsychological evaluation	
Total adjusted DRS-2 score	−2.06 (−4.25; −0.76)
Total DRS-2 < P5%, n (%)	27 (57.4%)

Abbreviations: CLD, chronic liver disease; DRS, Dementia Rating Scale; MELD, Model for End-Stage Liver Disease; ND, neurologic disease; OCM, oculomotor; P, percentile; SD, standard deviation.

**Table 2 biology-12-00804-t002:** Whole blood trace element levels (µg/L) in acquired hepatocerebral degeneration (AHD) patients (n = 51) and a control group of blood donors (n = 51).

Trace Element	AHD	Controls	*p*	Reference Values **
Manganese	14.6 (9.3–21.1)	6.0 (4.6–7.7)	<0.001	4.7–18.3
Arsenic	12.7 (9.9–20.2)	6.6 (2.5–10.8)	<0.001	<13
Boron	33.8 (25.1–51.3)	27.6 (20.8–35.5)	0.017	<100 ^1,2^
Cadmium	0.402 (0.205–0.558)	0.245 (0.139–0.424)	0.007	<5
Cobalt	0.331 (0.269–0.469)	0.295 (0.214–0.417)	0.176	<1
Cesium	3.71 (2.42–4.53)	3.80 (3.13–5.59)	0.368	n.a.
Copper	968 (785–1188)	855 (799–946)	0.052	500–1500 ^3^
Indium	0.066 (0.048–0.077)	0.074 (0.038–0.087)	0.567	n.a.
Lithium	1.57 (1.24–2.74)	0.38 (0.32–0.55)	<0.001	n.a.
Magnesium *	39.2 (32.2–45.5)	38.2 (35.8–42.5)	0.874	17–231 ^1,2^
Molybdenum	0.958 (0.748–1.26)	0.570 (0.420–0.720)	<0.001	0.6–4.0
Nickel	1.32 (1.06–1.62)	0.74 (0.16–1.44)	<0.001	<2 ^3^
Lead	27.3 (11.8–45.1)	10.7 (8.3–18.7)	<0.001	<50
Rubidium *	3.2 (2.6–4.3)	5.3 (4.6–5.9)	<0.001	n.a.
Selenium	128 (118–150)	166 (156–193)	<0.001	150–241
Strontium	18.9 (14.8–24.9)	14.5 (11.8–17.3)	<0.001	5–20 ^3^
Antimony	10.2 (3.9–13.9)	4.1 (3.8–4.5)	<0.001	<3
Thallium	0.075 (0.045–0.117)	0.035 (0.029–0.049)	<0.001	<2
Zinc *	4.96 (4.28–9.76)	5.79 (4.89–6.23)	0.579	4.4–8.6 ^3^
Cu/Zn ratio	0.180 (0.124–0.217)	0.155 (0.142–0.193)	0.932	n.a.
Cu/Se ratio	8.6 (7.2–10.6)	6.4 (5.4–7.2)	<0.001	n.a.

* Results in mg/L; ** reference values for whole blood, using ICP-MS, taken from Mayo Clinic Laboratories, except when it is otherwise indicated; ^1^ serum/plasma; ^2^ other technique than ICP-MS; ^3^ Labcorp (Laboratory Corporation of America); n.a.—reference values not available.

**Table 3 biology-12-00804-t003:** Whole blood trace element levels (µg/L) in acquired hepatocerebral degeneration (AHD) patients divided by increasing hepatic impairment according to Child–Pugh classification criteria.

	Child–Pugh A (n = 21)	Child–Pugh B (n = 20)	Child–Pugh C (n = 10)	*p* (A × B)	*p* (A × C)	*p* (B × C)
Manganese	15.2 (9.6–21.5)	11.2 (7.9–21.7)	14.5 (9.6–21.2)	0.306	1.000	0.422
Arsenic	10.4 (8.2–16.5)	13.9 (11.5–25.9)	15.1 (10.0–33.0)	0.035	0.147	0.914
Boron	27.3 (19.4–44.3)	34.2 (28.7–53.0)	41.8 (29.8–107.6)	0.098	0.039	0.373
Cadmium	0.307 (0.165–0.461)	0.422 (0.267–0.661)	0.406 (0.206–0.826)	0.055	0.150	0.914
Cobalt	0.299 (0.236–0.374)	0.324 (0.272–0.458)	0.466 (0.315–0.521)	0.386	0.010	0.109
Cesium	2.7 (2.2–4.4)	3.4 (2.3–4.0)	4.6 (3.6–5.1)	0.990	0.079	0.024
Copper	1016 (846–1183)	832 (778–1224)	889 (546–1137)	0.633	0.159	0.307
Indium	0.068 (0.060–0.075)	0.058 (0.044–0.076)	0.074 (0.046–0.86)	0.183	0.525	0.201
Lithium	1.40 (0.98–1.83)	1.62 (1.36–2.41)	3.20 (1.52–5.91)	0.051	0.031	0.143
Magnesium *	39.5 (32.9–45.4)	38.5 (31.9–45.5)	42.7 (31.6–46.0)	0.957	0.724	0.910
Molybdenum	0.861 (0.719–1.060)	0.978 (0.774–1.340)	1.100 (0.813–1.330)	0.225	0.175	0.820
Nickel	1.32 (0.96–1.52)	1.30 (1.02–1.77)	1.36 (1.20–1.60)	0.430	0.546	0.846
Lead	16.4 (10.8–29.9)	35.2 (14.5–44.0)	55.1 (13.3–83.5)	0.088	0.053	0.169
Rubidium *	3.3 (2.5–4.4)	3.0 (2.4–4.1)	3.4 (2.7–4.5)	0.494	0.787	0.475
Selenium	134 (128–168)	123 (115–138)	119 (115–135)	0.019	0.013	0.495
Strontium	17.3 (14.7–25.4)	21.5 (13.7–26.4)	21.8 (15.3–24.5)	0.510	0.755	0.681
Antimony	6.1 (4.0–13.0)	9.6 (4.0–14.2)	11.9 (3.1–14.3)	0.610	0.441	0.940
Thallium	0.053 (0.038–0.085)	0.078 (0.047–0.132)	0.087 (0.073–0.200)	0.111	0.010	0.282
Zinc *	5.26 (4.39–9.99)	4.79 (3.92–8.97)	4.91 (4.24–9.92)	0.328	0.755	0.771
Cu/Zn ratio	0.180 (0.114–0.224)	0.181 (0.142–0.218)	0.122 (0.112–0.220)	0.694	0.398	0.152
Cu/Se ratio	8.5 (6.7–10.8)	8.9 (8.2–10.3)	8.9 (6.1–10.7)	0.186	0.819	0.713

* Values in mg/L.

**Table 4 biology-12-00804-t004:** Spearman correlations between clinical variables and blood trace element levels in acquired hepatocerebral degeneration (AHD) patients.

	CLDd	MELD	Ammonia	AHDd	Li	B	Mg	Mn	Co	Ni	Cu	Zn	As	Se	Rb	Sr	Mo	Cd	In	Sb	Cs	Tl	Pb	Cu/Zn	Cu/Se
Age	0.117	−0.081	0.044	−0.069	0.076	0.116	0.046	−0.021	0.030	0.125	0.300 *	0.049	0.265 ^#^	0.158	−0.032	−0.037	0.247 ^#^	−0.058	0.175	0.036	0.064	−0.104	0.057	0.223	0.260 ^#^
CLDd		−0.006	−0.088	0.201	−0.019	−0.040	0.007	0.012	0.060	0.067	−0.061	0.071	0.008	−0.019	0.113	−0.035	0.017	−0.103	0.197	0.038	0.163	0.161	−0.270 ^#^	−0.103	−0.126
MELD			−0.195	−0.309 *	0.246 ^#^	0.139	−0.004	0.071	0.326 *	0.146	−0.201	−0.174	0.221	−0.297 *	−0.021	−0.079	0.195	0.369 **	−0.010	0.107	0.100	0.267 ^#^	0.142	0.025	−0.050
Ammonia				0.270 ^#^	−0.314 *	−0.190	0.025	0.287 *	−0.067	−0.011	−0.053	−0.115	−0.016	0.017	0.131	−0.164	−0.133	0.223	−0.232	−0.043	−0.185	−0.105	−0.086	0.111	−0.032
AHDd					−0.007	0.201	0.254 ^#^	−0.037	0.053	0.160	0.165	0.127	0.125	0.255 ^#^	0.200	0.153	−0.073	−0.269 ^#^	0.114	0.220	0.264 ^#^	0.081	−0.133	−0.190	−0.087
Li						0.551 **	0.282 *	−0.048	0.449 **	0.152	0.066	0.074	0.447 **	−0.081	0.080	0.309 *	0.566 **	0.163	0.191	0.341 *	0.613 **	0.179	0.168	−0.116	0.091
B							0.212	−0.130	0.309 *	0.165	0.151	0.160	0.533 **	0.145	0.095	0.372 **	0.453 **	−0.055	0.059	0.190	0.650 **	0.280 *	0.399 **	−0.241 ^#^	0.027
Mg								0.399 **	0.599 **	0.746 **	0.525 **	0.625 **	0.418 **	0.526 **	0.606 **	0.112	0.134	−0.017	0.487 **	0.779 **	0.431 **	0.361 *	0.379 **	−0.324 *	0.129
Mn									0.544 **	0.447 **	0.328 *	0.258 ^#^	0.068	0.220	0.408 **	0.117	0.112	0.171	0.294 *	0.340 *	−0.060	0.245 ^#^	0.059	−0.045	0.218
Co										0.654 **	0.302 *	0.338 *	0.377 **	0.104	0.471 **	0.198	0.382 **	0.236 ^#^	0.464 **	0.609 **	0.308 *	0.422 **	0.339 *	−0.192	0.230
Ni											0.420 **	0.494 **	0.378 **	0.275 ^#^	0.538 **	0.005	0.212	0.017	0.537 **	0.648**	0.202	0.518**	0.222	−0.277 *	0.153
Cu												0.451 **	0.296 *	0.566 **	0.327 *	0.119	0.249 ^#^	−0.116	0.394 **	0.334 *	0.174	0.046	0.243 ^#^	0.129	0.696 **
Zn													0.248 ^#^	0.638 **	0.642 **	−0.041	−0.114	0.000	0.477 **	0.488 **	0.394 **	0.236 ^#^	0.271 ^#^	−0.732 **	−0.042
As														0.281 *	0.145	0.335 *	0.347 *	−0.078	0.252 ^#^	0.383 **	0.334 *	0.441 **	0.352 *	−0.171	0.060
Se															0.335 *	0.015	−0.074	−0.140	0.273 ^#^	0.251 ^#^	0.320 *	0.076	0.192	−0.337 *	−0.111
Rb																−0.142	−0.101	0.126	0.309 *	0.671 **	0.440 **	0.367 **	0.179	−0.459 **	0.029
Sr																	0.304 *	−0.325 *	−0.006	0.017	0.111	−0.119	0.257 ^#^	0.025	0.140
Mo																		0.073	0.174	0.098	0.245 ^#^	0.063	0.122	0.161	0.399 **
Cd																			−0.175	0.068	0.060	−0.069	0.061	−0.026	−0.053
In																				0.492 **	0.164	0.194	−0.035	−0.203	0.237 ^#^
Sb																					0.350 *	0.465 **	0.175	−0.304 *	0.117
Cs																						0.251 ^#^	0.295 *	−0.409 **	−0.127
Tl																							0.176	−0.408 **	−0.170
Pb																								−0.137	0.189
Cu/Zn																									0.527 **

* *p* < 0.05; ** *p* < 0.01; ^#^
*p* ≥ 0.05 and <0.1. AHDd, acquired hepatocerebral disease duration; CLDd, chronic liver disease duration; MELD, Model for End-Stage Liver Disease.

**Table 5 biology-12-00804-t005:** Neurological assessment and blood trace element levels (μg/L) in acquired hepatocerebral degeneration (AHD) patients before and after liver transplantation (n = 6).

Neurological Evaluation	Before LT n (%)	After LT n (%)	*p*
Cognitive decline	5 (83.3)	2 (33.3)	0.083
Behavioral disorders	0	0	1.000
Mood disorders	0	0	1.000
Dysarthria	3 (50)	3 (50)	1.000
Parkinsonism	4 (66.7)	0	0.046
Tremor	6 (100)	5 (83.3)	0.317
Dystonia	0	0	1.000
Gait disorders	4 (66.7)	1 (16.7)	0.180
OCM dysfunction	1 (16.7)	0	0.317
Peripheral neuropathy	0	1 (16.7)	0.317
Pyramidal symptoms	2 (33.3)	1 (16.7)	0.564
Cerebellar syndrome	2 (33.3)	1 (16.7)	0.564
Neuropsychological evaluation	Before LT	After LT	*p*
Total adjusted DRS-2 score	−3.28 (−4.64; −1.24)	−2.11 (−3.20; −0.15)	0.068
Total DRS-2 < P 5%, n (%)	4 (66.7%)	4 (66.7%)	1.000
Trace elements	Before LT median (IQR)	After LT median (IQR)	*p*
Manganese	12.9 (7.50–18.5)	9.9 (4.4–14.3)	0.463
Arsenic	10.8 (7.72–19.2)	22.4 (12.3–31.8)	0.116
Boron	30.4 (23.0–98.7)	43.3 (21.2–66.5)	0.463
Cadmium	0.506 (0.466–0.702)	0.386 (0.222–0.750)	0.345
Cobalt	0.320 (0.246–0.458)	0.348 (0.178–0.412)	0.917
Cesium	4.1 (2.3–4.4)	4.1 (2.7–5.1)	0.463
Copper	904 (711–1147)	937 (848–1145)	0.753
Indium	0.060 (0.038–0.085)	0.056 (0.042–0.078)	0.752
Lithium	1.64 (1.15–3.03)	1.97 (0.67–3.06)	0.345
Magnesium *	34.0 (31.6–45.5)	32.7 (30.6–46.7)	0.600
Molybdenum	0.976 (0.958–1.26)	1.13 (0.972–1.47)	0.600
Nickel	1.17 (0.90–1.53)	1.16 (0.74–2.13)	0.600
Lead	11.0 (7.9–41.2)	20.9 (9.0–41.7)	0.046
Rubidium *	2.8 (2.7–3.9)	3.1 (2.6–4.4)	0.345
Selenium	127 (115–162)	168 (151–191)	0.028
Strontium	14.5 (8.43–21.6)	18.6 (11.3–23.6)	0.028
Antimony	4.5 (3.3–11.2)	2.5 (2.2–12.1)	0.345
Thallium	0.076 (0.047–0.217)	0.052 (0.044–0.083)	0.173
Zinc *	4.71 (3.91–10.7)	11.4 (10.4–11.8)	0.028
Cu/Zn ratio	0.142 (0.109–0.230)	0.088 (0.076–0.098)	0.028
Cu/Se ratio	8.3 (6.3–10.3)	5.7 (5.2–6.2)	0.046

* Values in mg/L. Abbreviations: DRS, Dementia Rating Scale; LT, liver transplantation; OCM, oculomotor; IQR, interquartile range.

## Data Availability

The data presented in this study are available on request from the corresponding author. The data are not publicly available due to privacy and ethical restrictions.

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
