# Peer review of "Trace Element Imbalances in Acquired Hepatocerebral Degeneration and Changes after Liver Transplant"

_biology, 2023, doi:10.3390/biology12060804_

Round 1

Reviewer 1 Report

The study was adequately designed. The manuscript is well and qualitatively written. However, it is necessary to clarify and correct certain parts of the manuscript.

Line 36-37: explain "male/female ratio: 2.64"; also line 152

Line 79-80: clarify the choice of analyzed elements

Line 100-101: The ethics clearance number must be in the manuscript

How do you explain the use of K2EDTA vacutainers, since you analyzed the magnesium? Please explain, because EDTA is a divalent ion chelator.

The conclusion must be more strongly written.

-

Author Response

Dear Editors,

We would like to thank the reviewers for their valuable suggestions, which we accept. We have answered directly to each question in the text below and highlighted the changes made to the article in the word file.

If anything else is necessary from our side, we are at your disposal.

With our best regards,

Henrique Nascimento

----------------------------------------------------------------------------------------

Reviewer 1

  • Line 36-37: explain "male/female ratio: 2.64"; also line 152.

Answer: we changed to percentage of males in order to make it clearer.

  • Line 79-80: clarify the choice of analyzed elements.

Answer: Naturally, we wanted to analyze the broadest possible panel of trace elements. But we exclude a priori the trace elements that: a) we know to be typically present in the blood at levels well below the detection limits that we can normally obtain with the equipment / analytical methodology we use; and b) those elements that we are unable to validate, due to accuracy issues, with our analytical equipment / methodology (e.g., Cr, V).

Routinely, in parallel with samples from patients and healthy subjects, we also analyze Certified Reference Materials / Quality Control (QC) samples (Seronorm™ Trace Elements Whole Blood, form SERO AS, Billingstad Norway, Level-1, Level-2 and Level-3, in this case) for analytical QC purposes (accuracy). Some elements are heavily interfered in ICP-MS (e.g. Cr or V), so special devices (e.g. collision/reaction cell) are needed to remove/reduce interferences and be able to determine their concentration accurately. Our equipment does not have such devices. Thus, in this article, we consider only those elements that passed the routine analytical QC.

  • Line 100-101: The ethics clearance number must be in the manuscript

Answer: we introduced the reference in the text.

  • How do you explain the use of K2EDTA Vacutainers, since you analyzed the magnesium? Please explain, because EDTA is a divalent ion chelator.

Answer: EDTA would be a problem if the determination of Mg in blood samples was done by the routine colorimetric method generally used in Clinical Laboratories, because EDTA could prevent the Mg from reacting with the color reagent (e.g., xylidyl blue). However, Mg was analyzed by ICP-MS. And with the ICP-MS technique, this problem does not occur. In ICP-MS, diluted blood samples are sent (nebulized) into very high temperature plasma (6 000 – 10 000 ºC). In plasma, elements are atomized, ionized and measured regardless of the chemical form in which they are present in the sample (whether as free ions, complexed or bound to proteins, etc.).

  • The conclusion must be more strongly written.

Answer: We changed the conclusion to be more incisive. However, we tried to keep it clear that further studies are need for better understanding some of the relations we reported in this paper.

Reviewer 2 Report

Thisn is a very interesting paper, showing novel information about trace element status and their shift after LT.

Minor methodical questions should be answered within the paper:

In INtroduction and material and methods sections a wide list bof elements is provided to be investigated in this paper. Among them the elements Al, Ag, Be, Bi, Sn and Te are mentioned, but no further concentration information about those elements is given. There is nothing about them in results section. Thus they should be removed from Introduction and M&M section. In case they were actually measured but below LOQ, this should be mentione together with rewspective LOQ but also with experimental settings.

In the § "Laboratory evaluation" the elements, but not the measured isotopes are listed. This is importance particularly for those elements, where some isotopes are interferred. Further all ICP-MS parameter have to be provided, including means for reduction or elimination of interferences. Was KED or DRC mode applied? and if so by which KED/DRC gases and parameter? If not, how could you eliminate the interferences?

BD-Vaccutainer are mentioned. Sb contamination is discussed in the paper, but other elements can be falsely elevated too (e.g. Al)

Author Response

Dear Editors,

We would like to thank the reviewers for their valuable suggestions, which we accept. We have answered directly to each question in the text below and highlighted the changes made to the article in the word file.

If anything else is necessary from our side, we are at your disposal.

With our best regards,

Henrique Nascimento

----------------------------------------------------------------------------------------

Reviewer 2

  • In Introduction and material and methods sections a wide list of elements is provided to be investigated in this paper. Among them the elements Al, Ag, Be, Bi, Sn and Te are mentioned, but no further concentration information about those elements is given. There is nothing about them in results section. Thus they should be removed from Introduction and M&M section. In case they were actually measured but below LOQ, this should be mentioned together with respective LOQ but also with experimental settings.

Answer:  There was an error in the original text. The phrase applies to all elements except Al. Actually, Al was not considered because it is difficult to control sample contamination (falsely elevated results), as shown with our quality control samples (please see our last answer). Reference to this element has been excluded from the entire manuscript.

The other elements (Ag, Be, Bi, Sn and Te), yes, they were analyzed but their levels in both groups with mostly below the LOD. This is relevant information if accompanied by the indication of LOD values. So, in the last paragraph of the fourth page we wrote: “A large percentage of Ag, Be, Bi, Sn and Te results were below the limit of detection (LOD) (respectively, 0.282, 0.031, 0.019, 0.131 and 0.157 µg/L), so these elements were excluded from the comparison between groups.”.

  • In the "Laboratory evaluation" the elements, but not the measured isotopes are listed. This is importance particularly for those elements, where some isotopes are interfered. Further all ICP-MS parameter have to be provided, including means for reduction or elimination of interferences. Was KED or DRC mode applied? and if so by which KED/DRC gases and parameter? If not, how could you eliminate the interferences?

Answer: The isotopes analyzed are now indicated in the section “Laboratory evaluation”, revised version. The ICP-MS instrument used does not have KED, DRC or any other interference elimination/attenuation system. However, the software allows using correction equations for some elements (e.g., As). Or, for example, in the case of Pb, adding the signals of different isotopes to increase sensitivity. As stated in response to reviewer #1 above, as a routine in our Laboratory, in parallel with the study samples, we always analyze quality control (QC) samples at the beginning, middle and end of each analytical series. In this study, we considered only those elements that passed the analytical QC.

The complete description of the analytical procedure and operational parameters of the ICP-MS instrument, including the elemental isotopes evaluated (including as internal standards), as well as the results obtained in the quality control can be found in our recent article (Azevedo, R.; Gennaro, D.; Duro, M.; Pinto, E.; Almeida, A. Further Evidence on Trace Element Imbalances in Haemodialysis Patients—Paired Analysis of Blood and Serum Samples. Nutrients 2023, 15, 1912. https://doi.org/10.3390/nu15081912, section 2.3. Laboratory Procedures). We have included this reference in the article, to which we refer the reader.

  • BD-Vacutainer are mentioned. Sb contamination is discussed in the paper, but other elements can be falsely elevated too (e.g. Al)

Answer: Sb is used as a catalyst in PET manufacturing and therefore contamination of blood samples collected in tubes made from this polymer may occur. This is a known fact, as discussed in the text. Except this, BD Vacutainer® Trace Element Tubes are widely used for trace element analysis and contamination with other elements has not been described. Contamination of samples with Al occurs very easily, not exactly due to the tubes but due to environmental conditions of sample processing, contamination of reagents, pipettes, etc. So, Al was not considered in the study.

Round 2

Reviewer 1 Report

The manuscript can be published.

-